# PROCESS REINFORCEMENT THROUGH IMPLICIT REWARDS

## ABSTRACT

Dense process rewards have proven a more effective alternative to the sparse outcome-level rewards in the inference-time scaling of large language models (LLMs), particularly in tasks requiring complex multi-step reasoning. While dense rewards also offer an appealing choice for the reinforcement learning (RL) of LLMs since their fine-grained rewards have the potential to address some inherent issues of outcome rewards, such as training efficiency and credit assignment, this potential remains largely unrealized. This can be primarily attributed to the challenges of training process reward models (PRMs) online, where collecting high-quality process labels is prohibitively expensive, making them particularly vulnerable to reward hacking. To address these challenges, we propose PRIME (**P**rocess **R**einforcement through **IM**plicit r**E**wards), which enables online PRM updates using only policy rollouts and outcome labels through *implicit process rewards*. PRIME combines well with various advantage functions and forgoes the dedicated reward model training phase that existing approaches require, substantially reducing the development overhead. We demonstrate PRIME's effectiveness on competitional math and coding. Starting from Qwen2.5-Math-7B-Base, PRIME achieves a 15.1% average improvement across several key reasoning benchmarks over the SFT model. Notably, our resulting model, Eurus-2-7B-PRIME, surpasses Qwen2.5-Math-7B-Instruct on seven reasoning benchmarks with 10% of its training data.

## 1 INTRODUCTION

Dense process rewards, which provide feedback at each intermediate step rather than only the whole trajectory, have proven effective in inference-time scaling of large language models (LLMs) on challenging reasoning tasks (Uesato et al., 2022; Lightman et al., 2023; Wang et al., 2023; Yuan et al., 2024b). On the training side, they also present superiority in the reinforcement learning (RL) of LLMs, particularly in improving training efficiency (Sutton & Barto, 2018) and credit assignment (Leike et al., 2018) compared with sparse outcome rewards. However, successful applications of dense rewards in RL for LLMs are limited (Setlur et al., 2024), as current industry-leading models primarily depend on verifiable outcome rewards and have not yet demonstrated meaningful progress with dense rewards (DeepSeek-AI et al., 2025; Team et al., 2025).

We identify the central challenge as *how to acquire and utilize high-quality dense rewards at scale*, which enables online process reward model (PRM) update efficiently. The reason is that, optimizing towards a static reward model eventually leads to overoptimization or reward hacking (Gao et al., 2022) due to distribution shift. Ideally, this can be solved by improving the reward model online (Leike et al., 2018). However, acquiring dense process labels for training is prohibitively more expensive. Existing methods either need to build complicated human annotation pipelines (Lightman et al., 2023) or rely on estimation-based methods, which require about $10\times$ more rollouts for each step than sampling only the response-level trajectories (Wang et al., 2023; Kazemnejad et al., 2024). Neither of them is scalable in online RL. Moreover, to the best of our knowledge, it remains underexplored how to incorporate dense rewards into RL for LLMs.

In this work, we propose Process Reinforcement through Implicit Rewards (PRIME), a scalable framework for enhancing reasoning capabilities via efficient reinforcement learning with dense token-level rewards. At its core, the framework employs recently proposed implicit process

reward modeling (Yuan et al., 2024b) to train dense reward models with only outcome-level labels. This enables PRIME to perform online learning of reward signals using only outcome labels on policy rollouts, thereby fundamentally mitigating reward hacking while maintaining the same computational cost as traditional outcome reward models (ORMs). Besides scalability, PRIME also (1) serves as a general method to fuse token-level dense rewards and sparse outcome rewards by calculating their returns separately before summing together, which is compatible with diverse RL algorithms (Williams, 1992; Kool et al., 2019; Shao et al., 2024; Ahmadian et al., 2024; Schulman et al., 2017); (2) eliminates the dedicated reward modeling stage, which is required by existing works, by simply initializing from the SFT model or even the base model (§ B.4). In summary, starting from one single language model, the PRIME framework can efficiently accomplish the generation of dense rewards, the initialization and updating of reward models, as well as the reinforcement learning (RL) training of the policy model.

In experiments, we train Qwen2.5-Math-7B-Base (Yang et al., 2024b) with PRIME after a lightweight SFT warmup stage. Compared to RL using outcome rewards only, PRIME achieves a $2.5\times$ sample efficiency gain and a $6.9\%$ performance improvements on challenging math problems. As shown in Figure 12, through PRIME, we successfully achieve substantial improvement on key mathematical reasoning benchmarks over the SFT model, leading to **16.7%** improvement on average, and over **20%** on AMC&AIME competitions. Our final model Eurus-2-7B-PRIME surpassed Qwen2.5-Math-7B-Instruct on five key mathematical benchmarks. Notably, this is achieved with only $10\%$ of the data used by Qwen-Math, as in Table 3.

Our analysis shows that updating the PRM online is key to the success of PRIME (§5.1). We also show that PRIME could generally boost various RL algorithms, including RLOO (Ahmadian et al., 2024), REINFORCE (Williams, 1992), PPO (Schulman et al., 2017), and GRPO (Shao et al., 2024) (§5.3). In terms of the design choices of advantage estimate, we observe that Implicit PRMs are better to be used as reward models than value models (§5.4).

## 2 REINFORCEMENT LEARNING FOR LLMS AND THE CHALLENGES OF INCOPORATING DENSE REWARDS

Reinforcement Learning (RL) aims to learn an optimal policy $\pi_\theta$ that maximizes the expected cumulative discounted reward, namely return, when interacting with an environment. In the context of autoregressive language modeling, state at step $t$ is the concatenation of prompt $\mathbf{x}$ and current response $\mathbf{y}_{<t}$, and the action is the $t$-th token or step $y_t$.

### 2.1 RL PRELIMINARIES FOR LLMS

**Policy Gradient.** Policy gradient is a fundamental algorithm that directly optimizes this objective. Central to this approach is the advantage function $A_t$, which quantifies how much better an action is compared to alternatives in a given state:

$$\nabla_\theta J(\theta) = \mathbb{E}_{\mathbf{x}\sim\mathcal{D},\mathbf{y}\sim\pi_\theta}\left[\sum_{t=0}^{T}\nabla_\theta \log \pi_\theta(y_t|\mathbf{y}_{<t})A_t\right] \tag{1}$$

where $(\mathbf{x}, \mathbf{y})$ represents a pair of input and output. $\mathbf{x}$ is omitted for brevity. In practice, the advantage function is implemented as cumulative discounted rewards subtracting a baseline:

$$A_t = \sum_{s=t}^{T}\gamma^{s-t}r(y_s) - b \tag{2}$$

$\gamma \in [0, 1]$ is a discount factor that optionally decays future rewards, and $r(y_s)$ is the reward provided by the environment at time step $s$ with $x$ and $\mathbf{y}_{<s}$ being omitted in conditions. Eq. 2 is the general formula of the Monte-Carlo (MC) advantage estimate, which indicates that, the high-quality and dense reward at each step is crucial for RL. Different choices of $b$ include, e.g. directly using values (Williams, 1992), group average of rewards (Shao et al., 2024), and leave-one-out average of rewards (Ahmadian et al., 2024; Kool et al., 2019).

**Value Models.** Though the MC estimate is unbiased, it suffers from high variance because of the reliance on all future actions and rewards, which can be random and noisy. Value models, which

predict expected accumulated rewards starting from a state, are adopted to help reduce the variance in advantage estimation, such as Generalized Advantage Estimation (GAE; Schulman et al., 2016): $A_t^{\text{GAE}(\gamma,\lambda)} = \sum_{s=0}^{\infty} (\gamma\lambda)^s \delta_{t+s}$, where $\delta_t = r(y_t) + \gamma V(\mathbf{y}_{<t+1}) - V(\mathbf{y}_{<t})$ is the temporal difference (TD) error (Sutton, 1988), $V$ is a value model, and $\lambda$ controls the bias-variance tradeoff in advantage estimation. PPO (Schulman et al., 2017) is a representative of such actor-critic algorithms that explicitly train a value model along with the policy.

**Reward Sparsity.** Although dense rewards can be naturally integrated into the advantage function through Eq. 2, unfortunately, only outcome reward models (ORMs) are available in most practices of LLMs, i.e., only the final token bears a meaningful reward while intermediate tokens receive no rewards (Rafailov et al., 2023; Shao et al., 2024; DeepSeek-AI et al., 2025). In this bandit setting, $r(y_t) = 0$ for $t < T$ while $r(y_T)$ can be non-zero, and Eq. 2 becomes $A = r(y_T) - b$. This formulation, while simpler, can suffer from reward sparsity issues as the policy receives feedback only at the end of the entire generation. This may (1) encourage spurious solutions with incorrect processes but correct answers, (2) largely reduce sample efficiency in training, and (3) encounter the credit assignment problem (Sutton & Barto, 2018). These drawbacks could be further amplified on complicated tasks, which require more thinking and execution steps, urging the need of dense rewards (Uesato et al., 2022; Lightman et al., 2023). Some may consider employing a value model to mitigate the problem, as it predicts values at every step $t$. However, previous work showed that value models may not be able to solve the reward sparsity issue effectively due to training challenges, despite the additional computation overhead (Shao et al., 2024; Ahmadian et al., 2024). We will also empirically validate this claim in §5.4.

## 2.2 Key Challenges in Scalable Dense Rewards

The way to mitigate the reward sparsity problem is to adopt dense reward models, namely PRMs, which score model responses over each token or step. However, it is usually infeasible in practice to incorporate dense rewards into online RL because of three critical challenges in implementation.

**C1. Process rewards are hard to define.** It is difficult to collect step-level labels since reasoning steps do not naturally occur in sequences. Although tokens are easily distinguishable, annotating labels for each token is too costly. Moreover, defining the absolute correctness of intermediate processes as dense rewards can be ambiguous, as some incorrect steps can also positively contribute to the final answer by pruning searching branches (OpenAI, 2024; DeepSeek-AI et al., 2025).

**C2. PRM online updates are not scalable.** It is crucial to prevent reward overoptimization or reward hacking, which requires the reward model or value model to be updated online along with the policy model (Schulman et al., 2017; Gao et al., 2022). However, training PRMs often requires extensive nuanced step-level annotation, which is infeasible in online RL training. Therefore, this brings about considerable scalability and generalization concerns in dense rewards for RL.

**C3. Explicit reward modeling brings extra cost.** Training reward models requires extensive annotation and broad data coverage to ensure a good balance between adaptability to the policy distribution and generalization to distribution shifts. Hence, the explicit training stage introduces a very costly data collection and an additional training overhead, especially for PRMs which typically require stepwise labels.

Notably, DeepSeek-AI et al. (2025) shares similar conclusions and thus is impeded from incorporating PRMs into large-scale RL training.

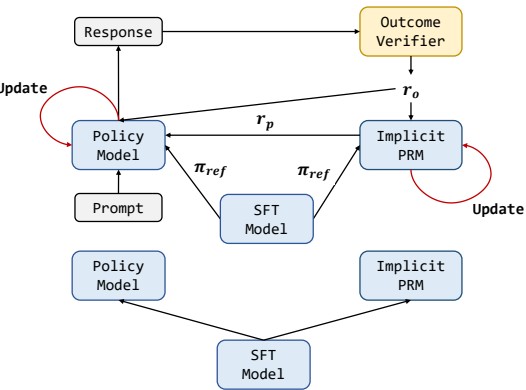

Figure 1: Illustration of PRIME. PRIME follows that (1) initialize policy model and the Implicit PRM both with the reference model; (2) sample multiple responses for each prompt and filter with output accuracy; (3) obtain implicit process rewards by the Implicit PRM and update it using cross-entropy (CE) loss; (4) compute advantage and policy loss then update the policy model.

---

**Algorithm 1** Process Reinforcement through Implicit Rewards (PRIME)

---

**Input** Language model $\pi_{\theta_{\text{init}}}$; outcome verifier $r_o$; dataset $\mathcal{D}$; sample number $K$; total iteration $N$.

1: Initialize policy model $\pi_\theta, \pi_{\theta_{\text{old}}} \leftarrow \pi_{\theta_{\text{init}}}$, implicit PRM and reference model $\pi_\phi, \pi_{\text{ref}} \leftarrow \pi_{\theta_{\text{init}}}$
2: **for** iteration = $1, \ldots, \text{N}$ **do**
3:     Sample batch of prompts $\mathcal{B} \sim \mathcal{D}$
4:     Generate $K$ responses: $\{\mathbf{y}^1, ..., \mathbf{y}^K\} \sim \pi_\theta(\cdot|\mathbf{x})$ for $\mathbf{x} \in \mathcal{B}$
5:     Compute outcome rewards: $r_o\left(\mathbf{y}^{1:K}\right)$
6:     Apply accuracy filter (§3.3) on all prompts: $\mathcal{T} \leftarrow \text{Filter}(\mathbf{x}, \mathbf{y}^{1:K}, r_o\left(\mathbf{y}^{1:K}\right))$ for $\mathbf{x} \in \mathcal{B}$
7:     Forward pass $\pi_\phi, \pi_{\text{ref}}$ on each $(\mathbf{x}, \mathbf{y}) \in \mathcal{T}$ to obatin implicit process reward $r_\phi(y_t)$ with Eq. 3
8:     Update Implicit PRM $\pi_\phi$ by CE loss on $(\mathbf{x}, \mathbf{y}, r_o(\mathbf{y})) \in \mathcal{T}$:

$$\mathcal{L}_{\text{CE}}(\phi) = -\mathbb{E}_{(\mathbf{x},\mathbf{y},r_o(\mathbf{y}))\sim\mathcal{T}}\left[r_o\left(\mathbf{y}\right) \cdot \log \sigma\left(r_\phi\left(\mathbf{y}\right)\right) + \left(1 - r_o\left(\mathbf{y}\right)\right) \cdot \log\left(1 - \sigma\left(r_\phi\left(\mathbf{y}\right)\right)\right)\right]$$

9:     Compute advantages $A$ with Eq. 5
10:     Update policy $\pi_\theta$ by PPO loss in Eq. 6
11:     Update old parameters: $\theta_{\text{old}} \leftarrow \theta$
12: **end for**

---

**Output** Optimized policy model $\pi_\theta$

---

## 3 PRIME

To address the above challenges, we propose PRIME, a scalable online RL method with dense rewards. The key insight of PRIME is to apply *implicit process rewards*, which are derivable from the Implicit PRM that is trained with only outcome labels (Yuan et al., 2024b). This property enables us to update the PRMs online to avoid reward hacking. We then design a flexible framework to incorporate implicit process rewards with outcome rewards into any kind of MC advantage estimate. PRIME is illustrated in Figure 1 and Algorithm 1. Next, we will detail the implicit process rewards (§3.1) and how we leverage them to calculate advantages (§3.2), and introduce other techniques we used (§3.3).

### 3.1 ENABLING SCALABLE REWARD UPDATE WITH IMPLICIT REWARD MODELING

We consider dense rewards from the Implicit PRM because of the scalability. In short, Implicit PRM enables training an ORM with outcome labels only while repurposing it as a PRM at inference. The training stage is the same as standard ORM pipelines, with the only difference being representing the reward as $r_\phi(\mathbf{y}) := \beta \log \frac{\pi_\phi(\mathbf{y})}{\pi_{\text{ref}}(\mathbf{y})}$, where $\pi_\phi$ is the RM and $\pi_{\text{ref}}$ is the reference model, both of which are causal LMs. At inference, the process rewards are obtained by:

$$r_\phi(y_t) := \beta \log \frac{\pi_\phi(y_t|\mathbf{y}_{<t})}{\pi_{\text{ref}}(y_t|\mathbf{y}_{<t})} \tag{3}$$

In PRIME, upon rollouts being generated and graded by the (ground truth) outcome verifier, we **update the Implicit PRM online with on-policy rollouts and outcome supervision** and then **calculate token-level dense rewards to estimate advantages**, which solves C1 and C2 mentioned in §2.2 respectively: (1) To prevent overoptimization and reward hacking, it is crucial to update reward models online. However, updating previous PRMs (Lightman et al., 2023) requires annotating step labels on the latest policy rollouts, which is neither efficient nor scalable during online RL. In contrast, the Implicit PRM only demands outcome labels to train due to its special reward representation, and thus it can be easily updated with policy rollouts and outcome labels or rewards, both of which have already been collected to update the policy model.

(2) Unlike common PRMs that produce only step-level rewards, the Implicit PRM provides more fine-grained *token-level* rewards at no additional cost. This addresses the ambiguity in identifying steps in LLM responses while not introducing extra overhead, making it easy to combine with any RL algorithms for advantage estimation. More discussions on Implicit PRMs are in § C.

## 3.2 ADVANTAGE ESTIMATION AND POLICY UPDATE

**Estimating advantages using Monte Carlo estimator with a leave-one-out baseline.** After obtaining token-level dense rewards, we calculate advantages based on either MC estimators or GAE. To determine the advantage function in PRIME, we compare GAE with several MC estimators, including REINFORCE (Williams, 1992), RLOO (Ahmadian et al., 2024), and GRPO (Shao et al., 2024). Experimental details and results can be found in §5.3.

We find that MC estimators, despite being simpler, are strong enough to produce stable results. Therefore, we choose MC estimate as our advantage function and despite PRIME being compatible with any baseline estimation approaches, we instantiate it with a leave-one-out baseline from $K$ samples (Ahmadian et al., 2024) in this paper, as it performs better in the experiments:

$$A^i = r_o(\mathbf{y}^i) - \frac{1}{K-1}\sum_{j \neq i} r_o(\mathbf{y}^j) \tag{4}$$

where $r_o(\mathbf{y}^i)$ denotes the reward of $i$-th response, $K$ is the number of samples for one prompt. The leave-one-out (LOO) baseline helps reduce variances.

More specifically, we use an Implicit PRM $\pi_\phi$ and an outcome verifier or reward model $r_o$. We calculate the return of implicit process rewards and outcome rewards separately if both are available, since directly mixing their values may lead to numerical instability (Shao et al., 2024). **For implicit process rewards**, we perform a three-step process to calculate return: (1) Use the averaged implicit process rewards to calculate the leave-one-out baseline; (2) Normalize the process reward at step $t$ by subtracting the baseline; (3) Calculate the discounted return for each response. **For outcome rewards**, we directly adopt LOO without any modification. Finally, the advantage is set to the combination of both returns:

$$A_t^i = \underbrace{\sum_{s=t}^{|\mathbf{y}^i|} \gamma^{s-t} \cdot \left[ r_\phi(y_s^i) - \frac{1}{K-1}\sum_{j \neq i} r_\phi(\mathbf{y}^j) \right]}_{\text{RLOO with implicit process rewards}} + \underbrace{r_o(\mathbf{y}^i) - \frac{1}{K-1}\sum_{j \neq i} r_o(\mathbf{y}^j)}_{\text{RLOO with outcome rewards}} \tag{5}$$

**Updating policy with PPO loss.** We adopt PPO clip surrogate loss for more stable policy updates:

$$L_{\text{CLIP}}(\theta) = \mathbb{E}_t \left[ \min\left( \frac{\pi_\theta(y_t|\mathbf{y}_{<t})}{\pi_{\theta_{\text{old}}}(y_t|\mathbf{y}_{<t})} A_t, \text{clip}\left( \frac{\pi_\theta(y_t|\mathbf{y}_{<t})}{\pi_{\theta_{\text{old}}}(y_t|\mathbf{y}_{<t})}, 1-\epsilon, 1+\epsilon \right) A_t \right) \right] \tag{6}$$

where $\epsilon$ is a clipping parameter. The loss prevents the updated policy from deviating too far from the original distribution, which is the prerequisite of importance sampling.

## 3.3 OTHER TECHNIQUES

**Initializing PRM with SFT/base model.** In practice, we find that the starting policy model itself serves as a decent initialization of PRM, bypassing the PRM training stage. This solves C3 in §2.2 and outperforms a dedicatedly trained PRM, as shown in § 5.1.

**Online Prompt Filtering.** As we sample multiple trajectories for each prompt, we introduce online prompt filtering which filters prompts within a certain accuracy range. This (1) preserves only the prompts within a certain median-level difficulty range (Yang et al., 2024b) and (2) balances data distribution for the Implicit PRM online training.

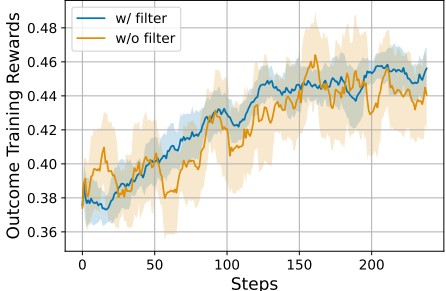

Figure 2: Effect of online prompt filtering.

We present the ablation study results in Figure 2 using RLOO with outcome rewards only, from which we can see that the online prompt filter largely lowers the variance of RL training.

**How PRIME addresses challenges in §2.2.** In summary, as illustrated in Figure 1 and Algorithm 1, PRIME adopts implicit process rewards for efficient PRM online update (C2), then integrates token-level dense rewards with outcome rewards in MC advantage estimate (C1). The PRMs are directly initialized from SFT or base models, which foregoes explicit reward modeling (C3).

Table 1: Detailed results of PRIME and RLOO w/ outcome verifier (OV). At the same 240 steps, the model trained by PRIME is generally better than the model trained by outcome rewards. We also reported avg@16 results in Table 12.

| Method | Step | AIME 2024 | AMC | MATH-500 | MinervaMath | OlympiadBench | LeetCode | LiveCodeBench | Avg. |
|---|---|---|---|---|---|---|---|---|---|
| GPT-4o | - | 9.3 | 45.8 | 76.4 | 36.8 | 43.3 | 58.9 | 48.8 | 45.6 |
| Llama-3.1-70B-Inst. | - | 20.0 | 37.3 | 65.0 | 37.1 | 30.5 | 35.0 | 34.4 | 37.0 |
| Qwen2.5-Math-7B-Inst. | - | 13.3 | 50.6 | 79.8 | 34.6 | 40.7 | 11.7 | 11.3 | 34.6 |
| Eurus-2-7B-SFT | 0 | 3.3 | 30.1 | 66.2 | 32.7 | 29.8 | 21.7 | 17.8 | 28.8 |
| RLOO w/ OV Only | 240 | **20.0** | 47.0 | 73.2 | 36.4 | 35.4 | 28.3 | 26.7 | 36.9 |
| | 80 | 20.0 | 41.0 | 68.2 | 38.2 | 37.0 | 26.7 | 26.6 | 36.8 |
| | 160 | 13.3 | 42.2 | 72.0 | 37.1 | 38.7 | 26.7 | 25.6 | 36.5 |
| Eurus-2-7B-PRIME | 240 | **20.0** | **50.6** | **78.2** | **39.3** | **40.3** | **31.1** | **27.5** | **41.0** |
| | 320 | 16.7 | 51.8 | 77.8 | 39.7 | 41.5 | 36.1 | 28.5 | 41.7 |
| | 592 | 26.7 | 57.8 | 79.2 | 38.6 | 42.1 | 33.3 | 28.6 | 43.9 |

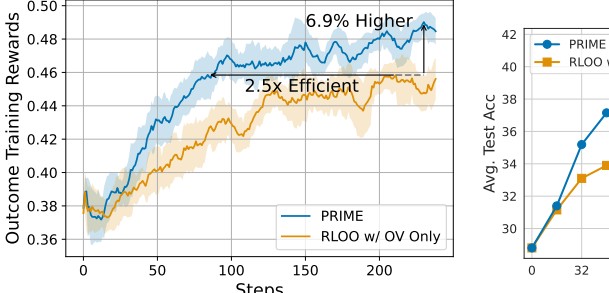
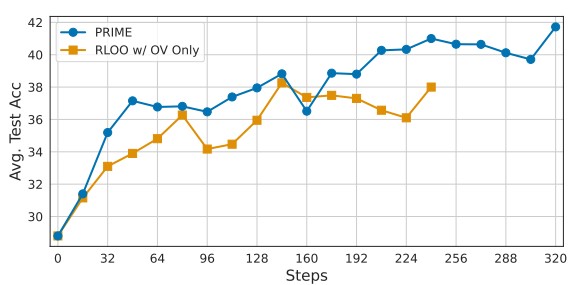

(a) Outcome training rewards (10-step moving).     (b) Test accuracy across different gradient steps.

Figure 3: **The effect of dense reward.** We compare PRIME and RLOO with outcome verifier (OV). PRIME leads to $2.5\times$ sample efficiency (wall clock as X axis can be found in Figure 17) and $6.9\%$ performance improvement. PRIME also substantially outperforms RLOO on downstream tasks.

## 4 EXPERIMENTS

We first perform supervised finetuning on the base model to get a starter model for RL. Please refer to § D for more details of this stage. All experiments are conducted on $8\times$A800 GPUs.

### 4.1 RL SETTINGS

**Rule-based Outcome Verifier.** Consistent with recent research that adopts an exact match with ground truth as unhackable rewards (Gao et al., 2024; Lambert et al., 2024; DeepSeek-AI et al., 2025), we define the rule-based ground truth outcome verifiers (OV) for math and coding as follows:

$$r_o^{\text{math}}(\mathbf{y}) = \begin{cases} 1, & \text{matched} \\ 0, & \text{otherwise} \end{cases} \qquad r_o^{\text{code}}(\mathbf{y}) = \frac{\sum \#\text{passes}}{\sum \#\text{test cases}}$$

**Hyperparameters.** We use veRL (Sheng et al., 2024) to conduct experiments. By default, we initialize the Implicit PRM with SFT model and retain the SFT model for reference logprobs. For hyperparameters, we use a constant $5 \times 10^{-7}$ learning rate together with AdamW optimizer for policy model, and use a $10^{-6}$ learning rate for PRMs. Both policy and PRMs use a batch size of 256 and micro batchsize of 8. The rollout stage collects 256 prompts and samples 4 responses for each prompt. We set $\beta = 0.05$ for PRM training. We set KL coefficient to 0 in all experiments.

**Evaluation Benchmarks.** We evaluate on 7 reasoning benchmarks, focusing on competition-level mathematics and programming tasks, including AIME 2024 (Li et al., 2024), AMC (Li et al., 2024), MATH-500 (Hendrycks et al., 2021b), Minerva Math (Lewkowycz et al., 2022), Olympiad-Bench (He et al., 2024), LeetCode (Guo et al., 2024), and LiveCodeBench (v2) (Jain et al., 2024).

### 4.2 MAIN RESULTS

As shown in Figure 12 and Table 1, Eurus-2-7B-PRIME achieves substantial improvements on key reasoning benchmarks over the SFT version of the model, leading to 15.1% improvement on av-

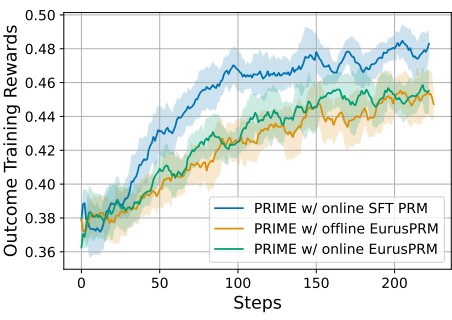 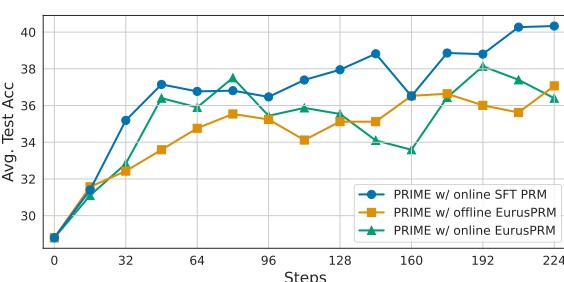

(a) Outcome training rewards (10-step moving).      (b) Test accuracy across different gradient steps.

Figure 4: **Comparison of different PRMs.** Online PRM initialized from SFT model achieved the best results. However, using PRMs trained on extra rollouts hurts the performance.

erage, and over 20% on AMC and AIME competitions. Besides, Eurus-2-7B-PRIME achieves 26.7% pass@1 on AIME 2024, surpassing GPT-4o, Llama-3.1-70B-Instruct, and Qwen2.5-Math-7B-Instruct, demonstrating its excellent reasoning ability. Additional results are in § B.

### 4.3 DENSE REWARDS V.S. SPARSE REWARDS

**Performance.** We first validate the effect of PRIME with dense rewards compared to RLOO with outcome rewards only. We train this model for 240 steps. For PRIME, we use the same setting and train the model for 592 steps. We plot the training rewards measured by the outcome verifier and test accuracy in Figure 3. **Compared with sparse reward, PRIME improves the final rewards by 6.9%, with lower variances.** On downstream tasks, PRIME also consistently outperforms OV only setup. Detailed results are listed in Table 1.

**Training Efficiency.** We provide detailed training time of each training step for PRIME and RLOO in Table 2. PRIME requires 24% more time cost compared with RLOO. However, as shown in Figure 3,

Table 2: Step-wise time cost of PRIME and RLOO.

| Time(s) | Rollout | Policy update | PRM update | Others | Sum |
|---|---|---|---|---|---|
| **PRIME** | 281.7 | 156.6 | 150.9 | 91.1 | 680.3 |
| **RLOO** | 282.4 | 157.9 | 0 | 90.4 | 530.7 |

PRIME only takes 40% of the training steps to achieve the same training rewards as RLOO. **This means PRIME would still be 2× more efficient than RLOO when estimated by training time.** Additionally, the single controller design of veRL requires no extra GPU memory since all other components (policy model, rollout engine) would be offloaded to CPU during PRM update.

## 5 ANALYSIS

### 5.1 DESIGN CHOICES FOR THE IMPLICIT PRM

The Implicit PRM is the key component of PRIME, and its design choices greatly affect RL. In this section, we explore two major factors: (1) the initialization model and (2) the update mechanism.

**SFT model initializes a good PRM.** Conventionally, we need to collect data to train RMs and PRMs, and then we can use them in RL. However, the Implicit PRM is a language model, so we can initialize it from any language model with the same tokenizer as the policy model. To investigate whether it is still necessary to train a PRM in advance, we conduct experiments with different PRM initialization strategies: with the SFT model itself and with a specially trained PRM. For the later one, we train EurusPRM from Eurus-2-7B-SFT with additional 500K data generated by Llama3.1 and Qwen2.5 series (data details in § E.5).

We report the experiment results in Figure 4. **Surprisingly, directly using Eurus-2-7B-SFT to initialize the**

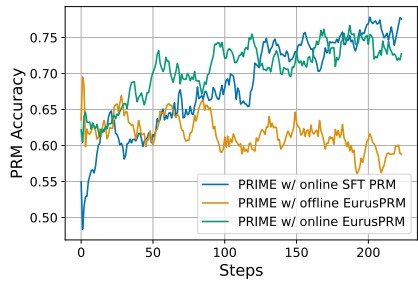

Figure 5: **Impact of PRM online update.** Offline PRM is gradually been overoptimized while online PRMs achieve higher accuracy during training.

**PRM greatly outperforms EurusPRM which was trained on more samples.** We conjecture that initializing policy model and PRM from the same model largely alleviates the distribution shift issue, as the PRM is only trained on the online rollouts from the policy model.

**Online PRM update is essential.** To verify the effect of online PRM update, we pair the correct and wrong samples and calculate the PRM prediction accuracy using $r_\phi(\mathbf{y})$. We report the PRM classification accuracy in Figure 5. The figure clearly shows that, **online update mitigates overoptimization and reward hacking.** The offline PRM, though starting with high accuracy, gradually drops during RL training procedure due to distribution shift. In contrast, online PRMs that are trained on policy rollouts show the reverse curve.

This is further validated with training rewards and downstream performance. To breakdown, Eurus-2-7B-SFT is both used as PRM initialization and the reference model in the main experiment, so the PRM is totally trained from scratch, which means the initial PRM outputs zero reward for all tokens. Therefore, Figure 3 also demonstrates the effect of online PRM update. For EurusPRM initialization, the online run outperforms the offline run as well in Figure 4.

## 5.2 SCALING PRIME WITH MORE COMPUTE

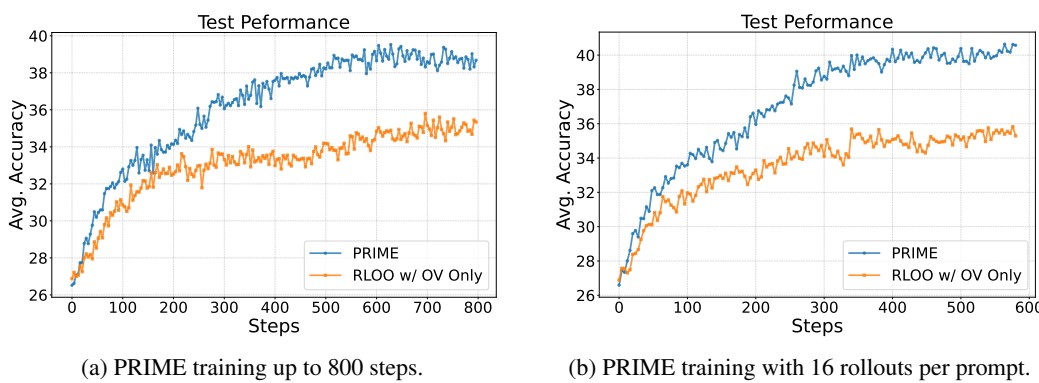

(a) PRIME training up to 800 steps.  (b) PRIME training with 16 rollouts per prompt.

Figure 6: RL training with more training steps (Left) and larger rollout numbers (Right).

To validate the scalability of PRIME with increased computational resources, we first conduct an extended training process. Specifically, we conduct RL training for 800 rollout steps (3200 gradient steps) with PRIME and RLOO with outcome-reward only. The training results, shown in Figure 6 (Left), reveal that throughout the training, PRIME consistently exhibits stable growth and outperforms the baseline with an improvement of 3.7%. Moreover, we increase the number of responses sampled for each prompt from 4 to 16. The results in Figure 6 (Right) show that PRIME achieves non-trivial improvement of approximately 4.4% compared to RLOO.

## 5.3 PRIME WITH OTHER RL ALGORITHMS

As we stated before, PRIME is equally applicable to other RL algorithms beyond RLOO. In this section, we implement PRIME with REINFORCE (Williams, 1992), GRPO (Shao et al., 2024), and PPO (Schulman et al., 2017). Similarly to RLOO, we only modify the advantage estimation functions and leave the clip surrogate loss unchanged. Detailed functions can be found in § B.3. From Figure 7 and Table 4, We show that PRIME boosts these algorithms on both efficiency and performance as it does with RLOO. PRIME contributes consistently regardless of the policy update method, making it a generic algorithm. It indicates that **PRIME is a general plug-in for almost any RL algorithm for LLM.**, which largely extends the use cases of PRIME.

## 5.4 VALUE OR REWARD, HOW TO USE THE IMPLICIT PRM?

Besides using process rewards to estimate returns, we can also employ the Implicit PRM to predict values for advantage estimation in Eq. 2. Therefore, we compare four variants of MC estimate to determine the best way to incorporate dense supervision. Recall that the Implicit PRM has

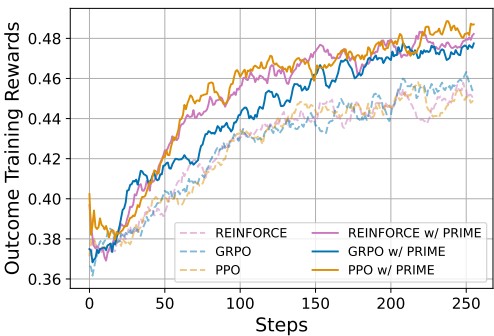 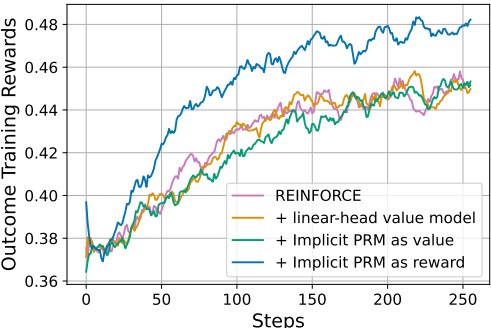

Figure 7: PRIME also generally benefits REIN-FORCE, GRPO, and PPO.

Figure 8: Comparison of value models and process reward models.

$v_\phi(\mathbf{y}_{<t+1}) = \sum_{i=1}^{t} \beta \log \frac{\pi_\phi(y_i|\mathbf{y}_{<i})}{\pi_{\text{ref}}(y_i|\mathbf{y}_{<i})}$ with the process reward being $r_\phi(y_t) = v_\phi(\mathbf{y}_{<t+1}) - v_\phi(\mathbf{y}_{<t})$, and we assume a ground-truth outcome verifier $r_o$, $\gamma = 1$, then we represent the variants as follows:

(1) REINFORCE: $A_t = r_o(\mathbf{y})$. (2) On top of (1), using **a linear-head value model** $V$ to calculate the baseline: $A_t = r_o(\mathbf{y}) - V(\mathbf{y}_{<t})$. This is the original PPO in Figure 7 as we set $\gamma = 1$ and $\lambda = 1$. (3) On top of (1), using **values from the Implicit PRM** to serve as the baseline: $A_t = r_o(\mathbf{y}) - v_\phi(\mathbf{y}_{<t})$. This is equivalent to PPO with its value model being replaced by values from the Implicit PRM when $\gamma = 1$ and $\lambda = 1$. (4) On top of (1), using **process rewards from the Implicit PRM** to calculate the return: $A_t = r_o(\mathbf{y}) + \sum_{s=t}^{T} r_\phi(y_s)$. This is exactly the REINFORCE w/ PRIME in Figure 7.

Figure 8 reports the results. Comparing PPO and REINFORCE, we find that an additional value model does not benefit policy performance. Notably, using rewards from the Implicit PRM to calculate returns, which is the default setting in PRIME, greatly outperforms all three baselines. This indicates that PRMs work better than value models in RL for LLMs. both two kinds of value models (PPO value model and Implicit PRM as value model) fall behind reward models.

## 6 RELATED WORK

**RL for LLM Reasoning.** In the area of LLMs, reinforcement learning has been widely used for aligning human preferences (Christiano et al., 2017; Ouyang et al., 2022; Cui et al., 2024), but the open-source community mostly adopt imitation learning methods (Yuan et al., 2024a; Yue et al., 2024; Wei et al., 2024; Liu et al., 2024) to enhance the reasoning capabilities of LLMs. Over the past few months, the paradigm gradually shifted. OpenAI o1 (Jaech et al., 2024) first showed the tremendous potential of large-scale RL for reasoning LLMs, and recent works have verified the scaling effect of RL with outcome rewards (DeepSeek-AI et al., 2025; Team et al., 2025). Meanwhile, the role of dense rewards in RL remains underexplored, serving the main focus of PRIME.

**Implicit Rewards.** Implicit rewards are broadly adopted in LLM alignment (Rafailov et al., 2023; Chen et al., 2024b; Azar et al., 2024; Ethayarajh et al., 2024; Rosset et al., 2024; Chen et al., 2024a). Rafailov et al. (2024) first showed that optimizing DPO objective learns a Q function implicitly. Zhou et al. (2024) utilized implicit rewards in PPO, and showed the effectiveness of dense implicit rewards. Yuan et al. (2024b) further extended the conclusion to any loss function optimizing Eq. 3.

## 7 CONCLUSION

As the fuel of LLMs, data, will be depleted in the near future, we are entering a new era of experience, which is exemplified by RL (Sutton, 2019). This work develops PRIME, which produces and leverages dense rewards in online RL for LLM reasoning. Throughout the experiments, we validate that PRIME (1) greatly benefits sample efficiency and policy performance, (2) is easy to use with minimum cost, and (3) is a general method that works with broad RL algorithms together.

ETHICS STATEMENT

This paper presents PRIME whose goal is to advance the field of reinforcement learning for LLMs. There are many potential societal consequences of our work, none of which we feel must be specifically highlighted here.

REPRODUCIBILITY STATEMENT

We have provided sufficient details to for reproduction, including algorithm pseudocode in Algorithm 1, experiment configurations and hyperparameters in Section 4 and Appendix. We have upload our code in Supplementary Material.

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

## A LIMITATIONS

Due to resource constraints, we only conducted experiments on models up to 32B. Besides the main experiments of PRIME, we ran fewer steps for other ablation experiments, while we conduct comparison under the same step number for fairness.

## B ADDITIONAL RESULTS

### B.1 REFERENCE MODEL CHOICE IS FLEXIBLE

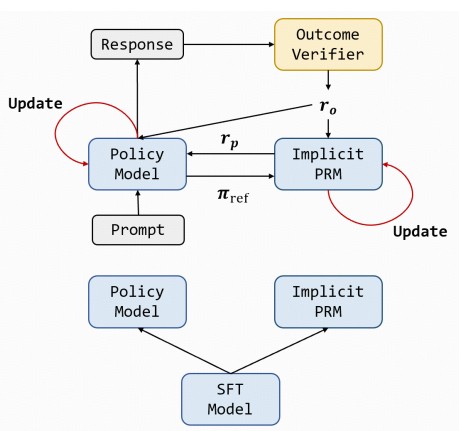 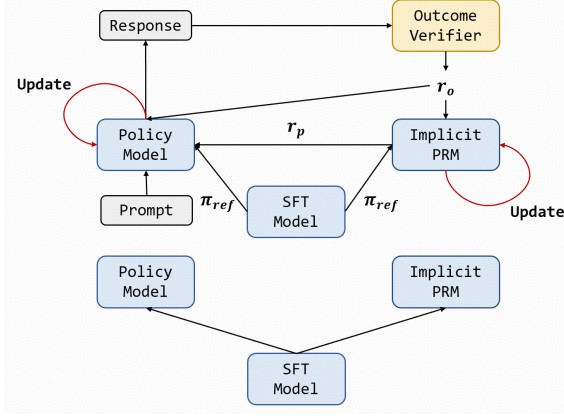

(a) Policy ref: We use the policy logprob as $\pi_{\text{ref}}$ for PRM.

(b) SFT ref: We retain the initial policy to provide $\pi_{\text{ref}}$ for PRM and KL.

Figure 9: Comparison of different reference policy implementations. One uses the running policy's old logprobs as reference (policy ref) while the other uses the initial SFT model as the reference model (SFT ref).

We implement two variants of our algorithms to explore the effect of reference model of implicit PRM, one using the initial SFT model as the reference model (SFT ref) while the other using the running policy's old logprobs as reference (policy ref), as shown in Figure 9a. The policy ref simply adopts the old logprob of the policy model as $\pi_{\text{ref}}$, while the SFT ref remains the initial SFT model for an additional $\pi_{\text{ref}}$ calculation. We compare their performance in this section.

From the training rewards in Figure 10, we find the two strategies are close and have pros and cons in different aspects: The Q value calculated by implicit PRM is the expectation under the distribution of the reference model. So the updating policy could natrually serve as the reference. On the other hand, KL divergence calculation is only allowed when the initial SFT model is retained.

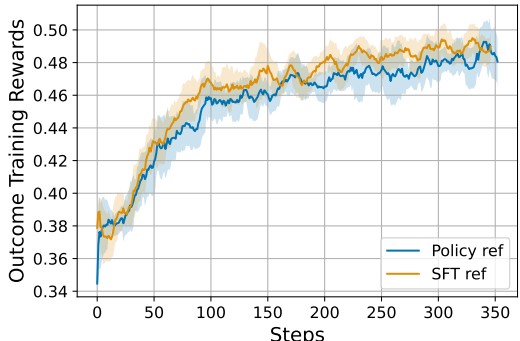

Figure 10: **Different reference model for PRM.** We compare two reference model selection strategies for PRIME. Using the policy model as reference and using the initial SFT model as reference. Their rewards are similar.

### B.2 SINGLE-FORWARD V.S. DOUBLE-FORWARD

Since our implicit PRM is concurrently updated in training, for each rollout stage, we can update the PRM before the policy model and use the updated PRM to re-calculate the process rewards, which

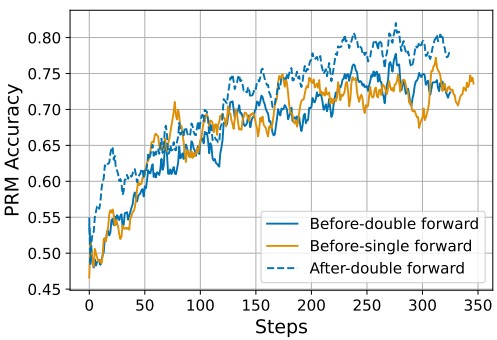 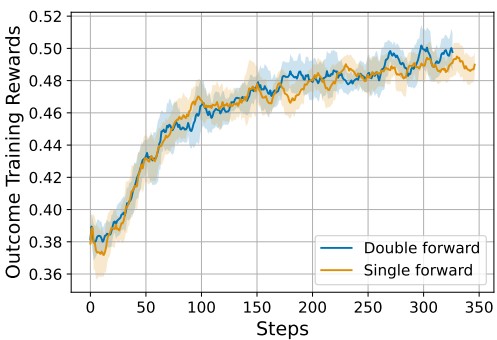

(a) PRM classification accuracy on training samples.  (b) Training outcome rewards.

Figure 11: **Single and double forward.** While double forward methods obtain higher accuracy after online update, the two variants achieve similar rewards during training.

Table 4: Testset results of different RL algorithms.

| Method | Step | AIME 2024 | AMC | MATH-500 | MinervaMath | OlympiadBench | LeetCode | LiveCodeBench | Avg. |
|---|---|---|---|---|---|---|---|---|---|
| **RLOO** | 240 | 20.0 | 47.0 | 73.2 | 36.4 | 35.4 | 28.3 | 26.7 | 36.9 |
| **RLOO w/ PRIME** | 240 | 20.0 | 50.6 | 78.2 | 39.3 | 40.3 | 31.1 | 27.5 | 41.0 |
| **REINFORCE** | 240 | 6.7 | 47.0 | 72.6 | 36.0 | 37.2 | 27.2 | 25.0 | 36.0 |
| **REINFORCE w/ PRIME** | 240 | 6.7 | 50.0 | 76.4 | 36.8 | 39.1 | 27.8 | 27.5 | 37.8 |
| **GRPO** | 240 | 10.0 | 44.6 | 73.2 | 37.5 | 36.6 | 25.0 | 25.8 | 36.1 |
| **GRPO w/ PRIME** | 240 | 16.7 | 47.0 | 75.0 | 34.9 | 38.2 | 28.9 | 23.9 | 37.8 |
| **PPO** | 240 | 10.0 | 41.0 | 73.6 | 36.0 | 36.3 | 28.3 | 25.7 | 35.8 |
| **PRIME as Value Model** | 240 | 16.7 | 44.6 | 72.6 | 34.6 | 35.7 | 27.8 | 24.6 | 36.6 |
| **PPO w/ PRIME** | 240 | 13.3 | 50.6 | 77.4 | 37.1 | 40.6 | 30.0 | 26.7 | 39.4 |

we call the double-forward setting. We investigate the impact of double-forward in both the training and test phases. Our default setting applies single-forward, which uses process rewards from old PRMs. We plot PRM accuracy on rollouts and training rewards in Figure 11.

Accordingly, we find that double-forward could increase PRM accuracy, but the training rewards remain close between the two methods.

## B.3 RESULTS OF DIFFERENT RL ALGORITHMS

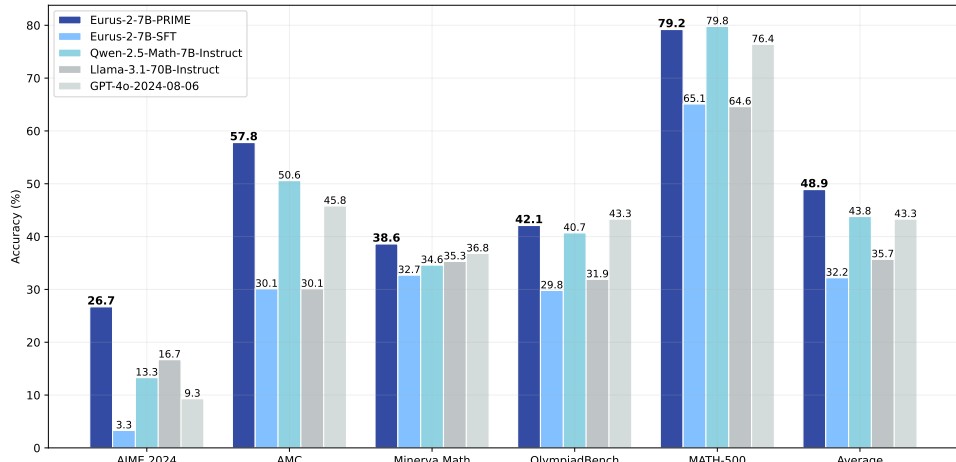

Figure 12: Overall math performance. Eurus-2-7B-PRIME excels at competition-level mathematics benchmarks, outperforming advanced math models and larger models. Notably, PRIME brings substantial performance gain (+16.7%) over Eurus-2-7B-SFT.

We ablate PRIME and different RL algorithms with their variants and find that the PRIME algorithm achieves the best performance for several reasons.

First of all, We compare different REINFORCE-like advantage estimators including RE-INFORCE, GRPO, and RLOO, toggling the existence of im-

Table 3: The comparison of resource requirements between Eurus-2-7B-PRIME and Qwen2.5-Math-7B-Instruct.

| Model | Eurus-2-7B-PRIME | Qwen2.5-Math-7B-Instruct |
|---|---|---|
| Base Model | Qwen2.5-Math-7B | Qwen2.5-Math-7B |
| SFT Data | 230K (open-source) | 2.5M (open-source & in-house) |
| RM Data | 0 | 618K (in-house) |
| RM | Eurus-2-7B-SFT | Qwen2.5-Math-RM (72B) |
| RL Data | 150K queries $\times$ 4 samples | 66K queries $\times$ 32 samples |

plicit process reward. To make different algorithms compatible with the compound of outcome verifier reward and process reward, we accordingly make adaptions similar to Eq. 5. For GRPO, we have

$$A_t^i = \underbrace{\frac{r_o\left(\mathbf{y}^i\right) - \text{mean}(r_o\left(\mathbf{y}^j\right))}{\text{std}(r_o\left(\mathbf{y}^j\right))}}_{\text{GRPO with outcome rewards}} + \underbrace{\sum_{s=t}^{|\mathbf{y}^i|} \gamma^{s-t} \cdot \left[ \frac{r_\phi(y_s^i) - \text{mean}\left(\frac{r_\phi(\mathbf{y}^j)}{|\mathbf{y}^j|}\right)}{\text{std}\left(\frac{r_\phi(\mathbf{y}^j)}{|\mathbf{y}^j|}\right)} \right]}_{\text{GRPO with implicit process rewards}}. \quad (7)$$

For REINFORCE, we have

$$A_t^i = \underbrace{r_o\left(\mathbf{y}^i\right)}_{\text{REINFORCE with outcome rewards}} + \underbrace{\sum_{s=t}^{|\mathbf{y}^i|} \gamma^{s-t} \cdot r_\phi(y_s^i)}_{\text{REINFORCE with implicit process rewards}}. \quad (8)$$

As shown in Table 4, PRIME contributes consistently regardless of the policy update method, making it a generic algorithm.

Moreover, the PPO variant of PRIME provides no performance gain, demonstrating that the additional computation cost from the critic model is redundant. This makes it possible to compensate for the expense of the process reward model by using REINFORCE-like algorithms with simpler advantage estimators.

Finally, we choose the best-performing RLOO as the advantage estimator in our algorithm.

### B.4 "ZERO" EXPERIMENTS

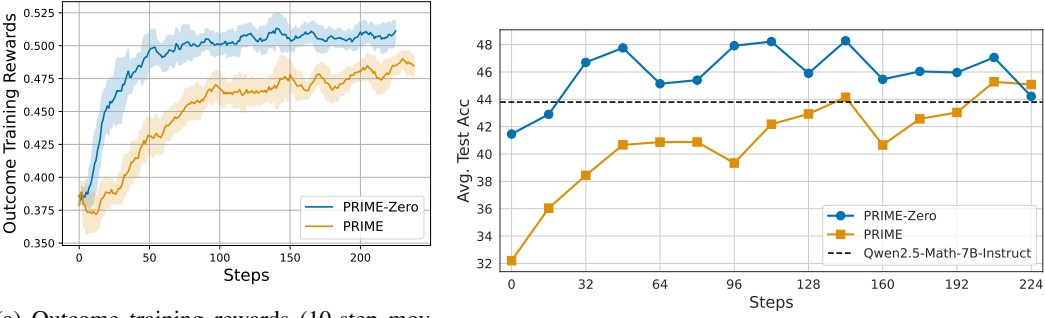

(a) Outcome training rewards (10-step moving).

(b) Math test accuracy across different gradient steps.

Figure 13: **"Zero" RL from Qwen2.5-Math-7B.** RL from the base model converges way faster than the SFT model, surpassing the instruct version within 32 steps.

DeepSeek-AI et al. (2025) proposed DeepSeek-R1-Zero, which is directly trained from a base model with reinforcement learning. To further investigate the "Zero" setting, we also perform RL from Qwen2.5-Math-7B-Base and Qwen2.5-32B-Base (Yang et al., 2024a), skipping the SFT phase. We present the experimental results in Figure 13 and Figure 14. The observations are as follows:

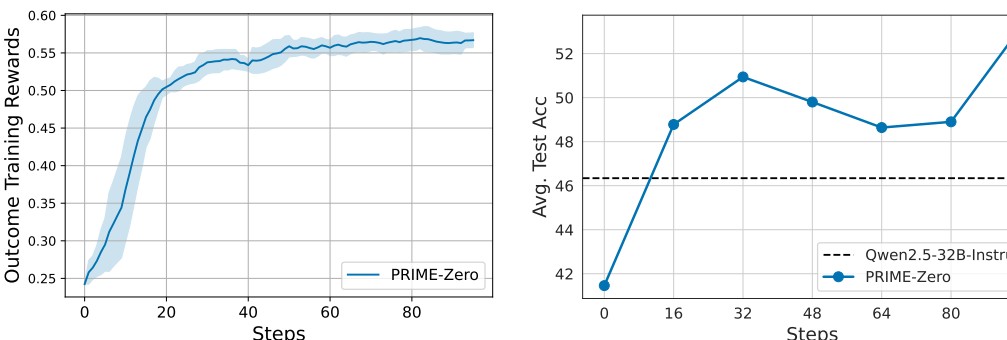

(a) Outcome training rewards (10-step moving).    (b) Math test accuracy across different gradient steps.

Figure 14: **"Zero" RL from Qwen2.5-32B-Base.** RL from a 32B base model shows more promising gain, surpassing the instruct version within 16 steps.

Table 5: Performance comparison of different reward models (Qwen2.5-7B-Base as policy model).

| Reward Model | AIME 24 | AIME 25 | AMC | MATH | Minerva | OlympiadBench | Average |
|---|---|---|---|---|---|---|---|
| **Qwen2.5-3B** | 10.7 | 4.8 | 44.0 | 73.2 | 26.1 | 33.0 | 32.0 |
| **Qwen2.5-7B** | 13.2 | 6.4 | 42.9 | 73.4 | 26.5 | 33.1 | 32.6 |
| **Qwen2.5-14B** | 10.8 | 4.8 | 44.1 | 73.2 | 25.4 | 32.7 | 31.8 |

(1) **RL from base model is suprisingly efficient and effective.** Comparing PRIME from Qwen2.5-Math-7B and Eurus-2-7B-SFT, the "Zero" setting converges much faster. This indicates that directly performing RL from a base model might be a strong alternative to the conventional SFT-RL pipeline.

(2) **Larger models benefit more.** Comparing 7B and 32B models, we see that the 32B model gains more on both training rewards and test performance. This is aligned with the conclusion in DeepSeek-AI et al. (2025).

(3) **Saturation could be a potential issue.** Although PRIME-Zero obtains impressive performance gain, we find it quickly saturated at a very early stage (about 50 steps), which hinders further improvement like in DeepSeek-AI et al. (2025). This is possibly attributed to the decrease of response diversity, and we leave this as future work.

## B.5    EFFECT OF REWARD MODEL SIZE

In the main experiments, we set the reward model to be of the same size as the policy model by default. To further investigate the influence of reward model capacity, we conduct comparative experiments by fixing the policy model as Qwen2.5-7B-Base and varying the reward model among Qwen2.5-3B-Base, Qwen2.5-7B-Base, and Qwen2.5-14B-Base, other settings are aligned with the main experiment. The results are summarized in Table 5. Overall, the results suggest that reward model size has limited influence: the 7B reward model achieves the best average performance, while larger (14B) or smaller (3B) reward models do not yield clear advantages.

## B.6    COMPARISON WITH VINEPPO

VinePPO (Kazemnejad et al., 2024) uses average return across trajectories to estimate the value within policy gradient update, instead of using value model in algorithm like PPO. We adopt the same setting as VinePPO, using RhoMath 1.1B (Lin et al., 2024) as the base model and MATH dataset for training. We reproduced PRIME and VinePPO with 8 A800-80G for 96 steps with the same hyperparameters. It reveals that

(1) **PRIME is 11x more efficient than VinePPO.** As shown in Figure 15, VinePPO takes 13.94 hours for training, while PRIME only needs 1.22 hours.

Table 6: Comparison between PRIME and VinePPO.

| Steps | 16 | 32 | 48 | 64 | 80 | 96 |
|---|---|---|---|---|---|---|
| **VinePPO Val Acc** (%) | 15.7 | 16.3 | 17.2 | 17.6 | 17.7 | 18.4 |
| **VinePPO Clock Time (Hours)** | 2.23 | 4.57 | 7.23 | 9.86 | 11.96 | 13.94 |
| **PRIME Val Acc** (%) | 16.4 | 16.8 | 17.5 | 18.1 | 18.7 | 18.8 |
| **PRIME Clock Time (Hours)** | 0.22 | 0.41 | 0.60 | 0.80 | 1.01 | 1.22 |

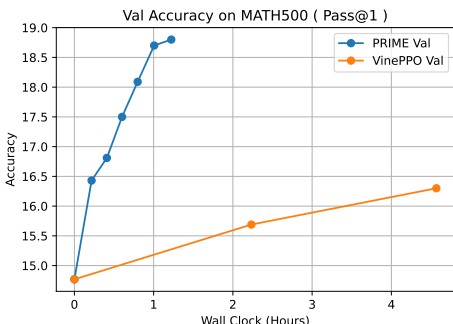 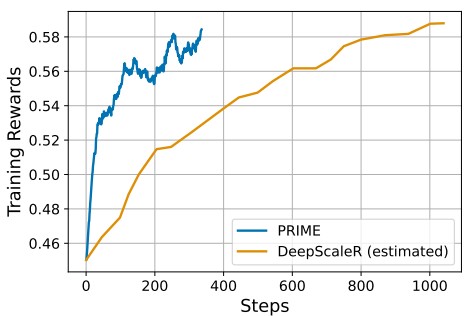

Figure 15: Validation accuracy curves of PRIME and VinePPO.

Figure 16: Training reward curves of PRIME and DeepScaleR.

(2) **PRIME also consistently outperforms VinePPO on the validation set.** As shown in the Table 6, the validation accuracy of PRIME is higher than VinePPO at each validation step.

## B.7 COMARISON WITH DEEPSCALER

DeepScaleR (Luo et al., 2025) introduces a three-stage training pipeline, with maximum allowed response length iteratively increased, from 8k (Stage 1) to 16k (Stage 2), finally to 24k (Stage 3). Through this iterative context lengthening, DeepScaleR achieve continuous performance gain. We run PRIME under the setting of DeepScaleR (base model, data, hyperparameters strictly follow the official settings). Due to resource limits, we only conduct the first (8k) stage training for 330 steps, yet PRIME achieves impressive results.

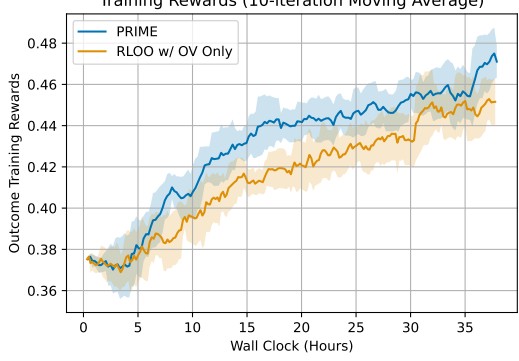

Figure 17: **The effect of dense reward.** We compared PRIME and RLOO with outcome verifier (OV). The figure depicts training reward curves across wall clock, revealing better sample efficiency of PRIME.

(1) **Model Performance.** Figure 16 presents the training accuracy curves of PRIME and Deep-ScaleR[1]. PRIME achieves comparable training accuracy within 330 steps, which is only 1/3 the steps of DeepScaleR stage 1 (1040 steps). On testsets, as shown in Table 7, PRIME consistently improves the performance of DeepSeek-R1-Distill-Qwen-1.5B by 3.1 points. This validates the effectiveness of PRIME on highly capable base models.

(2) **Efficiency.** PRIME consumed 446.7 A800 GPU hours for this experiment. In contrast, Deep-ScaleR consumed 3800 A100 GPU hours in total, and the first stage roughly required $\sim$600 GPU hours. This means PRIME is also 25% faster than DeepScaleR. Note that the advantage could be higher considering hardware differences (A800/A100).

(3) **Computation Overhead.** Moreover, due to the long response length of the distill model, the overhead would be attributed more to the generation phase, narrowing down the extra time PRIME brings to about 18%. This means that PRIME would be more suitable for long reasoning models.

Table 7: Comparison between PRIME and DeepScaleR.

| Model | Step | GPU Hour | AIME 2024 | MATH-500 | AMC | MinervaMath | OlympiadBench | Avg. |
|---|---|---|---|---|---|---|---|---|
| DeepScaleR-1.5B-Preview | 1750 | 3800 | 43.1 | 87.8 | 73.6 | 30.2 | 50.0 | 57.0 |
| DeepScaleR-1.5B-Stage1 | 1040 | $\sim$600 | 33.9 | - | - | - | - | - |
| DeepSeek-R1-Distill-Qwen-1.5B | - | - | 28.8 | 82.8 | 62.9 | 26.5 | 43.3 | 48.9 |
| PRIME-DeepScaleR-1.5B-Stage1 | 330 | 446.7 | 32.1 | 85.1 | 68.1 | 30.1 | 44.6 | 52.0 |

## C   DISCUSSION ON IMPLICIT PROCESS REWARD

### C.1   FORMULATION VALIDITY

As shown in Yuan et al. (2024b), implicit process reward is a parameterization of reward modeling. With such parameterization, the expectation of cumulative reward starting from the $y_t$ (i.e. q-value), $q_\phi^t(\mathbf{y}_{<t}, y_t) = \beta \log \mathbb{E}_{\pi_{\mathrm{ref}}(\mathbf{y}|\mathbf{y}_{\leq t})} e^{\frac{1}{\beta} r_\phi(\mathbf{y})}$ would have closed-formed solution $q_\phi^t(\mathbf{y}_{<t}, y_t) = \sum_{i=1}^t \beta \log \frac{\pi_\phi(y_i|\mathbf{y}_{<i})}{\pi_{\mathrm{ref}}(y_i|\mathbf{y}_{<i})}$. Therefore, despite the similarities between implicit process reward formulation to those of DPO (Rafailov et al., 2023; 2024), it is not derived from the optimal policy of entropy-regularized RL (Ziebart et al., 2008; Haarnoja et al., 2017).

### C.2   LOSS FUNCTION

Since $\pi_\phi$ and $\pi_{\mathrm{ref}}$ are language models which are inherently self-normalized, using cross-entropy loss brings about a minor issue that the minimum loss 0 cannot be reached. Therefore, the optimal solution would satisfy $\beta \log \frac{\pi_\phi^*(\mathbf{y})}{\pi_{\mathrm{ref}}(\mathbf{y})} = r_o + c$ rather than $\beta \log \frac{\pi_\phi^*(\mathbf{y})}{\pi_{\mathrm{ref}}(\mathbf{y})} = r_o$. This discrepancy would not affect RL, because PRIME uses relative reward, $r_\phi(\mathbf{y}^i) - \frac{1}{K-1} \sum_{j \neq i} r_\phi(\mathbf{y}^j)$, rather than the original reward from PRM. This means that even if the bias term is included, it would be canceled out in calculation since the bias term is only related with prompt $x$.

To solve this issue, we can simply eliminate this value term by using the DPO loss. We have a pilot experiment comparing DPO and CE loss, as shown in Table 8 and Figure 18. DPO and CE achieve similar results, and we chose CE for memory efficiency.

Table 8: Test accuracy of updating PRM with CE or DPO loss after training.

| Method | Step | AIME 2024 | AMC | MATH-500 | MinervaMath | OlympiadBench | Avg. |
|---|---|---|---|---|---|---|---|
| PRIME w. DPO loss | 96 | 7.7 | 39.3 | 66.2 | 17.3 | 31.3 | **32.4** |
| PRIME w. CE loss | 96 | 7.9 | 40.2 | 66.0 | 16.9 | 30.7 | **32.3** |

---

[1]Note that the provided training logs are broken according to this issue, so we estimated its training curve from the figure in the blog.

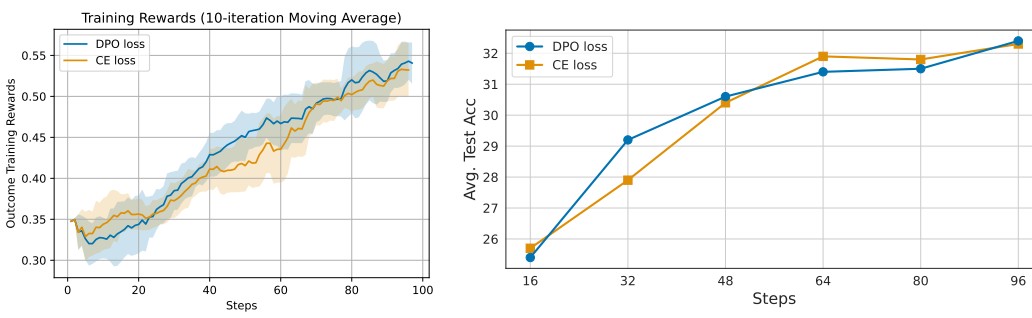

(a) Outcome training rewards.

(b) Math test accuracy across different gradient steps.

Figure 18: Outcome rewards and test accuracy of updating PRM with DPO or CE loss during training.

Table 9: Actions in action-centric chain-of-thought reasoning framework.

| Action Name | Description |
| --- | --- |
| ASSESS | Analyze current situation, identify key elements and goals |
| ADVANCE | Move forward with reasoning - calculate, conclude, or form hypothesis |
| VERIFY | Check accuracy of current approach, look for errors |
| SIMPLIFY | Break complex problems into simpler parts |
| SYNTHESIZE | Combine multiple pieces of information into complete solution |
| PIVOT | Change strategy when current approach isn't working |
| OUTPUT | Summarize thought process and present final answer |

## C.3 REWARD SHAPING

Another view of PRIME is potential-based reward shaping (Ng et al., 1999). If we view the q-value as the potential function, the process reward exactly satisfies the definition of shaping reward ($\gamma = 1$). PBRS does not affect the optimal policy, but does speed up learning, which is aligned with our results.

## D SFT DATA & TRAINING DETAILS

We first performed supervised finetuning for RL preparation. We focus on mathematical and coding problems in this paper. For models, we start with Qwen2.5-Math-7B-Base (Yang et al., 2024b) for its great mathematical capabilities.

**Action-centric chain-of-thought reasoning.** We apply imitation learning (supervised finetuning) as a warmup stage to teach models to learn certain reasoning patterns. To this end, we first design an action-centric chain-of-thought reasoning framework. Table 9 shows the actions in the action-centric chain-of-thought reasoning framework. When the model generates answers, it conducts multi-step reasoning and chooses one of the 7 actions at each step. The response begins with the ASSESS action and ends with the OUTPUT action.

**Construction of the SFT dataset.** To construct the SFT dataset, we collect reasoning instructions from several open-source datasets. It is noteworthy that we did not include many datasets with ground-truth answers in SFT, even though they are of higher quality. However, we reserve them for later RL training. The reason is that we aim to use different datasets for SFT and RL to diversify the exploration in RL, and we consider ground-truth more essential in RL than in SFT. For completion, we employ LLaMA-3.1-70B-Instruct to answer the instructions, with a system prompt requesting the model to perform an action-centric chain-of-thought. Table 10 summarizes the key statistics of the datasets used for SFT. The datasets span mathematics, coding, and biomedicine. We finally obtain 230K SFT data and the average response length is 1390 tokens.

**SFT Training.** During the SFT phase, we conduct full parameter fine-tuning with a learning rate of 1e-05, utilizing the AdamW optimizer alongside a cosine annealing learning rate schedule and a

Table 10: Data statistics of SFT data.

| Task | Dataset | Size | Avg. Response Length | Source |
|---|---|---|---|---|
| Math | MathInstruct-MATH (Yue et al., 2023) | 12715 | 964.01 | https://huggingface.co/datasets/TIGER-Lab/MathInstruct |
| | OpenMathIns-2-Aug_Math (Toshniwal et al., 2024) | 15086 | 1202.25 | https://huggingface.co/datasets/nvidia/OpenMathInstruct-2 |
| | Numina (Li et al., 2024) | 55845 | 1331.61 | https://huggingface.co/datasets/AI-MO/NuminaMath-CoT |
| | Reasoning-001 (SkunkworksAI, 2024) | 29831 | 1316.49 | https://huggingface.co/datasets/SkunkworksAI/reasoning-0.01 |
| Coding | Code-Feedback (Zheng et al., 2024) | 27663 | 1805.16 | https://huggingface.co/datasets/m-a-p/Code-Feedback |
| | Magicoder (Wei et al., 2024) | 24480 | 1828.72 | https://huggingface.co/datasets/ise-uiuc/Magicoder-Evol-Instruct-110K |
| | Magicoder-OSS (Wei et al., 2024) | 28980 | 1850.05 | https://huggingface.co/datasets/ise-uiuc/Magicoder-OSS-Instruct-75K |
| Biomedicine | UltraMedical_mc (Zhang et al., 2024) | 35163 | 891.06 | https://huggingface.co/datasets/TsinghuaC3I/UltraMedical |
| Total / Avg. | - | 229763 | 1390.75 | - |

warmup ratio of 0.1. The batch size was set to 96, with a fixed random seed of 42. The model was trained on 230K datasets for 3 epochs.

**SFT Results.** After finetuning, the performance of our SFT model is reported in Figure 12. Compared to baselines, Eurus-2-7B-SFT lags Qwen2.5-Math-7B-Instruct on all mathematics benchmarks.

## E  RL DATA PREPROCESSING

### E.1  RL DATA COLLECTION AND PREPROCESSING

We curate a high-quality RL training dataset of mathematics and coding problems with outcome verifiers (LaTeX answers for math and test cases for coding). For math, we source from NuminaMath-CoT (Li et al., 2024), which contains about 860K math problems. The problems span from Chinese high school mathematics to International Mathematical Olympiad competition questions. For coding, we source from APPS (Hendrycks et al., 2021a), CodeContests (Li et al., 2022), TACO (Li et al., 2023), and Codeforces[2]. To further increase data quality, we conduct detailed cleaning and filtering. Finally, we retain 457k math problems and 27k coding problems.

### E.2  DATA FILTERING AND QUESTION-TYPE CLASSIFICATION

The preprocessing pipeline employs a systematic rule-based approach to filter and classify mathematical problems to create a high-quality dataset with solvable problems, appropriate difficulty levels, and correct solutions. We exclude problems containing figures or diagrams since they require visual processing capabilities. We also remove proof questions due to difficulties in answer verification. Based on specific patterns, the remaining problems are classified into question-answering, multiple-choice, or fill-in-the-blank questions. Since fill-in-the-blank questions comprise less than 400 examples compared to the much larger set of multiple-choice questions, we focus solely on multiple-choice questions for further processing.

### E.3  CONVERTING TO DIRECT QUESTION-ANSWER FORMAT

We transform multiple-choice questions into a direct question-answer format through three sequential stages: rule-based filtering, LLM-based filtering, and LLM-based formatting.

We first identify and remove questions that inherently require multiple-choice options - specifically, those where comparing specific statements or properties is essential to the problem-solving process. These questions cannot be meaningfully converted to a direct question-answer format. The initial filtering employs simple rule-based pattern matching, searching for keywords like "following" and "statement" that typically indicate option-dependent problems.

Following the rule-based filtering, we employ Llama-3.1-8B-Instruct to perform a more nuanced classification of the remaining questions. Our pilot study revealed that while the LLM occasionally misclassifies questions, it tends to err on the conservative side - marking potentially convertible questions as requiring options rather than the reverse. Given our large dataset, we accepted this conservative approach to maintain quality.

---

[2]https://huggingface.co/datasets/MatrixStudio/Codeforces-Python-Submissions

Table 11: Data statistics of EurusPRM training dataset.

| Dataset | Generator Model | Num. Inst | Resp/Inst | Step-level/Response-level |
|---|---|---|---|---|
| UltraInteract | Llama-3.1-8B-Inst | 20177 | 8 | Response-level |
| | Llama-3.1-8B-Base | 13570 | 8 | Response-level |
| | Qwen2.5-72B-Inst | 4758 | 8 | Response-level |
| | Qwen2.5-Math-7B-Base | 25713 | 8 | Response-level |
| Numina-SynMath | Llama-3.1-8B-Inst | 4783 | 8 | Response-level |
| | Qwen2.5-Math-7B-Base | 5806 | 8 | Response-level |
| Numina-Olympiads | Llama-3.1-8B-Inst | 2909 | 8 | Response-level |
| | Qwen2.5-Math-7B-Base | 4739 | 8 | Response-level |

Table 12: Avg@16 results with temperature=0.3 of PRIME and RLOO w/ outcome verifier (OV).

| Method | Step | AIME 2024 | AMC |
|---|---|---|---|
| **Eurus-2-7B-SFT** | 0 | 4.4 | 21.4 |
| **RLOO w/ OV Only** | 240 | 15.4 | 43.8 |
| **Eurus-2-7B-PRIME** | 240 | 17.3 | 49.2 |
| **Eurus-2-7B-PRIME** | 592 | 24.2 | 54.5 |

For questions classified as convertible, we implement a two-phase reformatting process: 1) Question Reformatting: Removing choice indicators and restructuring the question to elicit direct answers. 2) Solution Reformatting: Converting multiple-choice solutions into step-by-step derivations, ensuring all final answers are presented in standard LaTeX boxed format. This systematic approach maintains mathematical rigor while creating a standardized format suitable for downstream applications.

### E.4 PROBLEM AND SOLUTION VALIDATION

The final stage involves merging all question-answer pairs and performing LLM-based comprehensive validation. We identify two key aspects in validation: solvability and correctness.

We leverage state-of-the-art mathematical reasoning models, including QwQ-32B-Preview (Team, 2024) and Qwen2.5-Math-72B-Instruct (Yang et al., 2024b), employing a self-consistency approach to determine problem solvability, and if solvable, verify the correctness of solutions provided in the original dataset.

To enhance validation accuracy, we first analyzed sample problems to identify characteristics of solvable and unsolvable cases and created synthetic unsolvable problems featuring missing conditions or logical contradictions. Based on these samples, we developed specialized prompts to improve the models' ability to distinguish solvability. Each problem undergoes five independent validation attempts, where the LLM: 1) Provides step-by-step solutions using LaTeX formatting. 2) Identifies unsolvability due to missing conditions or logical contradictions. 3) Generates complete reasoning traces for solvable problems. 4) Presents final answers in standardized LaTeX boxed format (\boxed{...}). 5) Document any impediments to solution completion.

We evaluate two key consistency measures across multiple validation attempts: 1) Status Consistency: agreement on problem solvability. 2) Answer Consistency: consistency of solutions across different attempts and agreement between generated solutions and ground truth. The final dataset retains only problems that demonstrate consistent solvability across validation attempts, agreement in solutions across multiple attempts, and alignment with ground truth answers. This rigorous validation process ensures the resulting dataset comprises well-defined, solvable problems with verified, accurate solutions.

### E.5 PRM DATA

The dataset statistics of training EurusPRM are shown in Table 11.

