# OpenReview forum: "Process Reinforcement through Implicit Rewards"
_ICLR.cc/2026/Conference — Submitted to ICLR 2026_

### Official Review · Reviewer_PFTD · 2025-10-15

**Soundness:** 2
**Presentation:** 3
**Contribution:** 2
**Rating:** 2
**Confidence:** 5

**Summary:**

This paper introduces a method for directly initializing process reward models from base or SFT models. These models are then incorporated into the reinforcement learning process. The authors also introduce an online update for the process reward models using cross-entropy loss. They evaluate their training pipeline on mathematics and programming tasks.

**Strengths:**

- This paper is clearly written and easy to follow.

- It addresses the important challenge of annotation-efficient process reward modeling in RL.

- The authors propose PRIME, a method that eliminates the need for a dedicated reward modeling stage by simply initializing the process reward model from the SFT or base model.

**Weaknesses:**

- **Limited Novelty**

There is a major concern about the novelty of the proposed methods for addressing the practical challenge of incorporating dense rewards into online RL, as outlined in section 2.2. The methods introduced in this paper appear to have been well studied in prior research. For instance, for Challenge 1, previous work [1,2,3] has already provided well-defined and theoretically sound token-level rewards. For Challenge 2, prior research such as [4] has adopted an online update strategy for reward modeling using MCTS and rule-based verification. For Challenge 3, existing studies such as [5] have established a theoretical foundation for deriving token-level rewards from pre-trained or SFT LLMs, which can then serve as training-free reward models for RL.

- **Lack of Clarification of Core Objective**

There is another major concern about the clarity of Equation 5 in the paper (in "RLOO with implicit process rewards"). The rationale for subtracting the sequence-level baseline $\frac{1}{K-1}\sum_{j\neq i}r_\phi(\mathbf{y}^j)$ from the token-level reward $r_\phi(y^i_s)$ is unclear.  Given that a token-level advantage function evaluates an action relative to other actions at the same state, using a baseline computed over entire sequences (with non-identical states) seems theoretically unsound. Furthermore, the definition of $r_\phi(\mathbf{y}^j)$ is ambiguous; I assume it represents  $\beta\log\frac{\pi_\phi(\mathbf{y}^j)}{\pi_\text{ref}(\mathbf{y}^j)}$, but this should be explicitly stated.

- **Lack of Justification for the CE Loss**

The paper provides insufficient theoretical and empirical justification for using CE loss to update the PRIME model at Algorithm 1. A more rigorous explanation is needed to clarify why this specific loss function is appropriate for the proposed framework.

- **Limited Experimental Baselines**

The experimental comparisons are limited. The main results only compare PRIME against an outcome verifier, while several highly relevant baselines [2,3,5,6] are missing. Furthermore, other related work [7,8] has also explored incorporating process reward models into RL. The omission of key comparisons raises concerns about the claimed effectiveness of the proposed framework.

[1] From r to Q: Your Language Model is Secretly a Q-Function, COLM 2024.

[2] Discriminative Policy Optimization for Token-Level Reward Models, ICML 2025.

[3] DPO Meets PPO: Reinforced Token Optimization for RLHF, ICML 2025.

[4] ReST-MCTS*: LLM Self-Training via Process Reward Guided Tree Search, NuerIPS 2024.

[5] Generalist Reward Models: Found Inside Large Language Models, Arxiv 2025.

[6] Preference-Grounded Token-Level Guidance for Language Model Fine-Tuning, NeurIPS 2023.

[7] Dense Reward for Free in Reinforcement Learning from Human Feedback, ICML 2024.

[8] Let's Verify Step by Step, Arxiv 2023.

**Questions:**

- Question about Equation 3. Why is this defined as a process reward? I think the authors should refer to the conclusion from some previous work (e.g., [1]) and provide further explanation in the paper.

- This work shares great similarity with previous work [2]. More discussion is needed to clarify this in the Related Work section.

[1] From r to Q: Your Language Model is Secretly a Q-Function, COLM 2024.

[2] Free Process Rewards without Process Labels, ICML 2025.

---

### Official Review · Reviewer_v7hP · 2025-10-21

**Soundness:** 3
**Presentation:** 2
**Contribution:** 3
**Rating:** 8
**Confidence:** 3

**Summary:**

The paper builds on the idea of implicit process rewards from Yuan et al (2024), which is a DPO-like transformation of outcome rewards into per-token process rewards through a DPO-like function $\log \sigma(\beta \log \frac{\pi_{\theta}(y_t | y_{<t}}{\pi_{ref}(y_t | y_{<t})})$ for "positive" examples (correct outcome) and $\log (1-\sigma(\beta \log \frac{\pi_{\theta}(y_t | y_{<t}}{\pi_{ref}(y_t | y_{<t})}))$ for negative samples (incorrect outcome).

This process reward is then converted into an advantage for PPO training, yielding improvements in overall performance and in efficiency, all in mathematical reasoning. Experiments are based mainly on Eurus-2-7b (which I think is just a fine-tuned variant of Qwen2.5-math-7b-Inst? unclear), and in comparison with RLOO (leave-one-out advantage estimation).

Additional experiments quantify the effect of online updates (figure 4 and 5), training efficiency (table 2), alternative RL algorithms (fig 7), and additional rollouts (fig 6).


Yuan et al 2024: https://arxiv.org/abs/2412.01981

**Strengths:**

The method seems relatively straightforward to implement, and seems to yield significant improvements on mathematical reasoning tasks.

The supplementary experiments comprehensively evaluate several aspects of the application of the method.

**Weaknesses:**

Many of the claimed contributions seem like they are really attributable to Yuan et al 2024 rather than this submission. While the paper cites Yuan et al repeatedly, it could be clearer about what specifically is novel in this submission.

Despite the extensive discussion of the motivation for the approach in 2.2, I'm still left uncertain as to why implicit process rewards work, given that there is no new information about which parts of the rollout were actually impactful. I think it would probably be possible to say more about this, though that may be out of scope for this paper.

**Questions:**

- Sorry if I missed it, but what is Eurus-2-7b-sft and how does it relate to Qwen?
- What specifically is the contribution of this paper wrt Yuan et al 2024?

---

### Official Review · Reviewer_CxT8 · 2025-10-26

**Soundness:** 2
**Presentation:** 3
**Contribution:** 2
**Rating:** 2
**Confidence:** 5

**Summary:**

The paper “Process Reinforcement through Implicit Rewards (PRIME)” proposes a scalable reinforcement learning framework for large language models (LLMs) that uses implicit process rewards instead of sparse outcome rewards. PRIME updates process reward models online using only outcome labels, eliminating costly step-level annotations and reducing reward hacking. It integrates token-level dense and outcome rewards into standard RL algorithms like PPO, REINFORCE, and RLOO. Experiments on math and coding benchmarks show significant gains—up to 15% improvement.

**Strengths:**

1. The paper is clearly written, well structured, and effectively communicates its methods, results, and insights.

2.  PRIME is practically designed, requiring no separate reward model training or step-level labeling. It can  straightforward to integrate into existing LLM training pipelines

**Weaknesses:**

1. The experiments focus mainly on mathematical reasoning and coding, which, although challenging, represent a narrow set of structured tasks.

2.  The proposed method only evaluated on Qwen based method. However, a few of works have been pointout that  qwen based method have serious test data leak problem.

Wu, Mingqi, et al. "Reasoning or memorization? unreliable results of reinforcement learning due to data contamination." arXiv preprint arXiv:2507.10532 (2025).

3. Lots of implicit reward formulation have been proposed from various perspective. This paper  omits discussion on possible bias or divergence between implicit and true process rewards.

**Questions:**

none

---

### Official Review · Reviewer_sryj · 2025-11-01

**Soundness:** 3
**Presentation:** 3
**Contribution:** 3
**Rating:** 6
**Confidence:** 4

**Summary:**

This paper presents PRIME, a novel reinforcement learning framework for LLMs that derives dense process-level rewards implicitly from outcome-level supervision. Instead of relying on costly, explicitly annotated process reward models, PRIME learns token-level feedback signals online by contrasting the current policy with a reference model, effectively bridging the gap between sparse and dense reward formulations. The proposed approach integrates seamlessly with existing RL algorithms and demonstrates strong empirical results across a variety of reasoning benchmarks. Overall, the method is elegant, efficient, and empirically validated.

**Strengths:**

The paper is well-organized, with clear motivation, algorithmic description, and empirical validation.
The implicit reward formulation is also conceptually sound and well-motivated, allowing dense process feedback without explicit annotations.
Moreover, PRIME consistently improves accuracy and sample efficiency across multiple reasoning benchmarks, often surpassing larger or more heavily trained baselines.
The method introduces minimal complexity and can be readily integrated into standard RL pipelines, making it both accessible and impactful for the community.
The experimental evaluation is also comprehensive.
The results are consistently strong and demonstrate both the effectiveness and robustness of the proposed method.

**Weaknesses:**

I did not find any major issues in the paper, but I still have a few questions that I hope the authors could clarify.

First, the authors mention that the proposed method can mitigate reward hacking. However, in domains such as math reasoning, where predefined correctness rules may already provide reliable outcome-based rewards, the advantages of implicit process rewards are less evident.
I think this paper would benefit from a more detailed discussion or empirical analysis on whether the proposed approach is indeed more effective in more general or less structured tasks.

The second concern is somewhat related, regarding generalization.
The experiments focus mainly on reasoning tasks, where both the task structure and the verifier are well defined. It would strengthen the paper to discuss or test PRIME’s potential applicability to open-ended or weakly structured domains (e.g., dialogue generation or summarization), where defining correct outcomes is more subjective.
I would be happy if the authors engage in a discussion on these points, and I am open to revising my score based on discussions.

**Questions:**

Please refer to the concerns in Weaknesses.

---

### Meta-Review · Area_Chair_FXAS · 2026-01-06

**Summary:**

The main concerns raised by the reviewers are as follows.
1. **Novelty (CxT8, v7hP and PFTD)**. Implicit reward has been extensively studied in the literature.
2. **Experiments (sryj, CxT8 and PFTD)**. It lacks benchmarks and baselines. Moreover, the Qwen back-born may have data leakage issues on certain benchmarks.
3. **Clarity (PFTD)**. The clarity and derivation of the implicit reward objective requires justification.

**Reviewer Concerns:**

The concerns are not addressed, as rebuttal is missing.

**Reviewer Scores:**

Reviewers are hard to change their score, as rebuttal is missing.

---

### Decision · Program_Chairs · 2026-01-26

Reject